# A machine learning approach to evaluate the state of hypertension care coverage: From 2016 STEPs survey in Iran

Hamed Tavolinejad[1,2], Shahin Roshani[1,3], Negar Rezaei[1,4], Erfan Ghasemi[1], Moein Yoosefi[1], Nazila Rezaei[1], Azin Ghamari[1], Sarvenaz Shahin[1], Sina Azadnajafabad[1], Mohammad-Reza Malekpour[1], Mohammad-Mahdi Rashidi[1], Farshad Farzadfar[1,4]*

1 Non-Communicable Diseases Research Center, Endocrinology and Metabolism Population Sciences Institute, Tehran University of Medical Sciences, Tehran, Iran, 2 Tehran Heart Center, Cardiovascular Diseases Research Institute, Tehran University of Medical Sciences, Tehran, Iran, 3 Netherlands Cancer Institute, Amsterdam, Netherlands, 4 Endocrinology and Metabolism Research Center, Endocrinology and Metabolism Clinical Sciences Institute, Tehran University of Medical Sciences, Tehran, Iran

* f–farzadfar@tums.ac.ir

## Abstract

### Background

The increasing burden of hypertension in low- to middle-income countries necessitates the assessment of care coverage to monitor progress and guide future policies. This study uses an ensemble learning approach to evaluate hypertension care coverage in a nationally representative Iranian survey.

### Methods

The data source was the cross-sectional 2016 Iranian STEPwise approach to risk factor surveillance (STEPs). Hypertension was based on blood pressure $\geq$140/90 mmHg, reported use of anti-hypertensive medications, or a previous hypertension diagnosis. The four steps of care were screening (irrespective of blood pressure value), diagnosis, treatment, and control. The proportion of patients reaching each step was calculated, and a random forest model was used to identify features associated with progression to each step. After model optimization, the six most important variables at each step were considered to demonstrate population-based marginal effects.

### Results

The total number of participants was 30541 (52.3% female, median age: 42 years). Overall, 9420 (30.8%) had hypertension, among which 89.7% had screening, 62.3% received diagnosis, 49.3% were treated, and 7.9% achieved control. The random forest model indicated that younger age, male sex, lower wealth, and being unmarried/divorced were consistently associated with a lower probability of receiving care in different levels. Dyslipidemia was associated with reaching diagnosis and treatment steps; however, patients with other

**Data Availability Statement:** The STEPS 2016 study is a project launched by the Iranian Ministry of Health and Medical Education of Iran (MOHME), who owns the rights to the dataset. The authors are

not legally allowed to publicly share the data on behalf of MOHME. However, interested and qualified researchers may contact the Non-Communicable Diseases Research Center (www.ncdrc.net; email address: ncdrc.sc@gmail.com; ncdrc.epid@gmail.com) to access the datasets of the STEPS 2016 study. The aggregated level data and reports are freely accessible via https://nih.tums.ac.ir/Show/Item/453?AspxAutoDetectCookieSupport=1. Independent researchers can use the following URLs to download: STEPs 2016 questionnaire: https://nih.tums.ac.ir/UpFiles/Documents/593d64f1-601a-4ae4-81dd-ab002117d186.pdf An aggregated report of STEPs 2016: https://nih.tums.ac.ir/UpFiles/Documents/34d5ee8d-864e-46df-be72-83de3178f833.pdf Guide to STEPs 2016 data: https://nih.tums.ac.ir/upfiles/documents/315067249.pdf A sample of the STEPs 2016 dataset: http://nihr.tums.ac.ir/upfiles/documents/311771543.xlsx.

**Funding:** This study was funded by Ministry of Health and Medical Education and National Institute for Health Research (grant number: 241-93259).

**Competing interests:** The authors have declared that no competing interests exist.

cardiovascular comorbidities were not likely to receive more intensive blood pressure management.

## Conclusion

Hypertension care was mostly missing the treatment and control stages. The random forest model identified features associated with receiving care, indicating opportunities to improve effective coverage.

## Introduction

Success in controlling communicable diseases, population growth, and aging have led to a demographic and epidemiologic shift in low- and middle-income countries (LMICs) [1]. As a result, the health-related burden of non-communicable diseases (NCDs) has become one of the most significant social and economic challenges facing LMICs towards sustainable development [1, 2]. The health-care systems of LMICs struggle in making the necessary adaptations since NCDs require longitudinal, patient-centered, and multilevel care [3]. Hypertension is a leading NCDs risk factor which can lead to mortality and morbidity [4], and over the past decades, the burden of hypertension has shifted to LMICs with an increase in the prevalence of high blood pressure (BP) [4, 5].

Assessment of health-care system performance and coverage for NCDs is essential to guide public health policies and succeed in reducing risk factors. Care cascade models are used to assess coverage and gaps in care for chronic infectious diseases such as human immunodeficiency virus and latent tuberculosis infection [6, 7]. Regarding NCDs, a number of studies have performed similar care cascade analyses [8, 9]. Thus far, an evaluation of national-level care for NCDs and their risk factors has not been reported from Iran. In this context, we used ensemble learning to evaluate the state of hypertension care. Such methods can be particularly useful to derive meaningful inferences from large datasets [10], which means data mining is a superior method for evaluating NCDs care. In this study, we aimed to discover the associations of receiving appropriate interventions through different stages of hypertension care by using the random forest model in a large nationally representative data.

## Materials and methods

### Data source

This analysis was performed on the 2016 STEPwise approach to risk factor Surveillance (STEPs) data from Iran. STEPs is a population-based, large-scale, cross-sectional study aiming to monitor NCDs based on the STEPs framework developed by the World Health Organization (WHO) [11]. The design of 2016 STEPs survey is further described elsewhere [12]. The STEPs data was deemed appropriate for this analysis since it samples a wide variety of communities, encompasses a broad age range, and employs a standardized method.

### Ethical considerations

The 2016 STEPs study complied with the latest edition of the declaration of Helsinki and was approved by the ethics committee of the National Institute for Medical Research Development (NIMAD Approval ID: IR.NIMAD.REC.1394.032). Participants received a detailed explanation of the rationale and objectives of the study and provided written informed consent before inclusion.

## Study population

Patients with hypertension, defined as the presence of either (I) systolic BP (SBP)≥140 mmHg or diastolic BP (DBP)≥90 mmHg; (II) ever using medications for hypertension; (III) or a previously diagnosed hypertension by a health-care provider (HCP) were included [13]. The 2016 STEPs anthropometry section required three BP measurements at three-minute intervals. If all three values were available, the mean of the latter two measurements was used. In participants with two readings, the first was discarded and the second measurement entered the dataset. An algorithmic description of hypertension definition is presented in S1 File/*Methods*.

## Steps of care

We determined steps as the proportion of hypertensive patients who fulfill a set of criteria. The first step (designated as "Screening") determined if the patient's BP had ever been measured by an HCP. It should be mentioned that fulfillment of screening was directly based on history of BP measurement by HCP, irrespective of the presence of hypertension, and without considering the measured BP value. The second step ("Diagnosis") was defined as ever receiving hypertension diagnosis by an HCP. BP equal to or higher than 140/90 mmHg, as defined in our hypertension population, usually indicates pharmacological treatment; hence the third step ("Treatment") was ever receiving anti-hypertensive medications. Reaching the fourth step ("Control") required having SBP<130 mmHg and DBP<80 mmHg [14]. The thresholds for defining hypertension and reaching control were selected to best reflect the state of care in Iran. These conventional cut-offs are widely used in LMICs [15], and enable comparison with other available data from these countries [8, 15]. Essentially, each step was a prerequisite for the next one. We defined the outcome as reaching steps of hypertension care and looked for characteristics associated with each level. Further details of the care cascade definition are available in S1 File.

## Associated features

We examined the association of receiving care with demographic characteristics (age, sex, marital status), socio-economic features (rural/urban residency, education level, wealth index, occupation, being the head of household, insurance coverage), and comorbidities (history of cardiovascular disease, smoking, dyslipidemia, diabetes mellitus, and body mass index [BMI] category). Education level was based on the number of years spent in school or university and included primary schooling, secondary education, academic education. Wealth index was previously defined for STEPs [16], and is based on quintiles derived from a Principal Component Analysis of the family assets. Insurance coverage levels were based on different health-insurance plans available in the country. BMI levels were categorized to underweight (BMI≤17.5), normal (17.5≤BMI<25), overweight (25≤BMI<30), obese (30≤BMI<35), and morbid obese (BMI≥35).

## Statistical modeling

Random forest is an ensemble model that handles categorical variables without the need to transform them to binary forms, and when it is well optimized through a proper resampling process, it provides appropriate and competitive predictive accuracy compared to other algorithms [17]. Moreover, optimization of hyper-parameters is highly efficient in random forests due to independence of trees, which leads to easy parallelization of the tree fitting process, and the unique availability of out of bag validation, which makes the validation process less time consuming, enabling the use of time and computational power to expand hyper-parameters

space for hyper-parameter tuning and obtain an even better form of the model. Indeed, there may be other algorithms that could achieve a slightly better accuracy for our analysis, but we chose random forest based on the above-mentioned considerations and the characteristics of our data.

Hyper-parameters are involved in various machine learning algorithms to control their complexity. More complex models have less bias but may have too much variability due to overfitting, while less complex models can be too shallow and have more biased results. In this sense, hyper-parameters must be tuned through an appropriate validation process to create a balance in algorithms bias-variance trade-off and deliver generalizable results. To control the complexity of our random forest model and avoid overfitting, models' performances were evaluated using accuracy as loss function for combinations of the hyper-parameters—mtry (sample size of predictors) = 2, 6, 10; minimum node size = 1, 5, 10; sample fraction of observations = 25%, 50%, 75%, 100%—in out of bag validation procedure with 50 resamples to ensure the generalizability of the validation results. Gini splitting rule was fixed in the validation process. In pre-processing stages, removing zero/near zero variance variables, bag-imputation of missing values, and Synthetic Minority Oversampling Technique (SMOTE) sampling [18] to limit the class imbalance effect of response variable, were used during the resampling process to avoid data leakage phenomenon.

After obtaining the final optimized random forest model, specific model-agnostic interpretation tools were used. Permutation-based variable importance [19, 20], with measuring the change in loss function after permutation of the targeted predictor, was used to rank variables so that bigger changes indicate more important variables. Subsequently, ranked variables were divided into quartiles and Partial Dependence Plot (PDP) [21, 22] were drawn to clearly demonstrate the population-based marginal effects for the most important quartile (4th quartile) of ranked variables. Importantly, existing interacting/confounding effects were taken into account by the model. Higher-dimension interactions of features were evaluated by partial dependence plots. It should be noted that dashed lines in PDPs do not indicate continuity between levels of categorical variables, and they were only drawn to facilitate visualizing the changes in PDPs. All procedures were done using R (Version: 3.6.1) and RStudio (Version: 1.2.1335).

## Results

The analysis included 30541 participants (52.3% female; median age: 42). The response rate reached 98.4% in STEPs 2016. According to our definition, 9420 (30.8%) individuals were hypertensive at the time of the survey, among whom 89.7% had ever had BP screening, 62.3% had received appropriate diagnosis, 49.3% had been treated for hypertension, and 7.9% had achieved BP control before the study. The characteristics of participants at each level of the care cascade are summarized in Table 1.

Associated variables were sorted according to their levels of importance in prediction of reaching care steps (results of hyper-parameter tuning for model optimization are presented in the S1 File/*Hyperparameter tuning*). The most important features emerging as good classifiers in the care continuum were age, sex, occupation, education, wealth index, marital status, being the head of household, and dyslipidemia.

Age was an important predictor in hypertension care, demonstrating the highest importance in screening, diagnosis, and treatment, as well as the second highest importance in hypertension control. In each of the four steps, older age was consistently associated with a higher likelihood of reaching higher levels. The shapes of PDPs support this interpretation, and with increasing age, the mean predicted probabilities (MPP) increased in all steps (Figs 1–

**Table 1. Population characteristics in each step of the care cascade.**

| | Hypertensive patients (n = 9420) | Screened (n = 8451) | Diagnosed (n = 5866) | Treated (n = 4643) | Controlled (n = 747) |
|---|---|---|---|---|---|
| **Demographic features** | | | | | |
| Age, years | | | | | |
| <45 | 2245 (23.83%) | 1811 (21.43%) | 982 (16.74%) | 463 (9.97%) | 66 (8.84%) |
| [45,56] | 2323 (24.66%) | 2077 (24.58%) | 1370 (23.35%) | 1047 (22.55%) | 155 (20.75%) |
| [56,66] | 2327 (24.70%) | 2185 (25.85%) | 1631 (27.80%) | 1403 (30.22%) | 219 (29.32%) |
| ≥66 | 2525 (26.80%) | 2378 (28.14%) | 1883 (32.10%) | 1730 (37.26%) | 307 (41.10%) |
| Female sex | 5209 (55.30%) | 4832 (57.18%) | 3601 (61.39%) | 2891 (62.27%) | 456 (61.04%) |
| Marital status | | | | | |
| Unmarried | 471 (5.03%) | 338 (4.02%) | 131 (2.25%) | 59 (1.28%) | 8 (1.08%) |
| Married | 7482 (79.88%) | 6739 (80.20%) | 4645 (79.72%) | 3612 (78.40%) | 569 (76.48%) |
| Divorced/separated | 163 (1.74%) | 141 (1.68%) | 92 (1.58%) | 70 (1.52%) | 15 (2.02%) |
| Widow/widower | 1251 (13.36%) | 1185 (14.10%) | 959 (16.46%) | 866 (18.80%) | 152 (20.43%) |
| Household head | 5101 (54.40%) | 4549 (54.07%) | 3076 (52.74%) | 2495 (54.13%) | 407 (54.70%) |
| **Socio-economic features** | | | | | |
| Area of residence | | | | | |
| Urban | 6514 (69.15%) | 5863 (69.38%) | 4082 (69.59%) | 3280 (70.64%) | 545 (72.96%) |
| Rural | 2906 (30.85%) | 2588 (30.62%) | 1784 (30.41%) | 1363 (29.36%) | 202 (27.04%) |
| Education | | | | | |
| Primary schooling | 3807 (41.84%) | 3504 (42.89%) | 2638 (46.68%) | 2259 (50.72%) | 341 (46.97%) |
| Secondary education | 2932 (32.23%) | 2581 (31.60%) | 1731 (30.63%) | 1306 (29.32%) | 236 (32.51%) |
| Academic education | 2359 (25.93%) | 2084 (25.51%) | 1282 (22.69%) | 889 (19.96%) | 149 (20.52%) |
| Wealth index | | | | | |
| Very low | 2072 (22.45%) | 1812 (21.89%) | 1312 (22.85%) | 1065 (23.47%) | 156 (21.34%) |
| Low | 1979 (21.44%) | 1743 (21.06%) | 1216 (21.18%) | 969 (21.36%) | 148 (20.25%) |
| Medium | 1829 (19.82%) | 1664 (20.10%) | 1142 (19.89%) | 907 (19.99%) | 137 (18.74%) |
| High | 1749 (18.95%) | 1579 (19.08%) | 1057 (18.41%) | 814 (17.94%) | 148 (20.25%) |
| Very high | 1601 (17.35%) | 1479 (17.87%) | 1015 (17.68%) | 782 (17.24%) | 142 (19.43%) |
| Occupation | | | | | |
| White-collar clerk | 546 (5.82%) | 490 (5.83%) | 304 (5.21%) | 208 (4.51%) | 26 (3.49%) |
| Blue-collar worker | 348 (3.71%) | 275 (3.27%) | 147 (2.52%) | 100 (2.17%) | 17 (2.28%) |
| Self-employed | 1834 (19.56%) | 1535 (18.25%) | 912 (15.62%) | 633 (13.72%) | 90 (12.08%) |
| Volunteer/conscript | 69 (0.74%) | 58 (0.69%) | 29 (0.50%) | 20 (0.43%) | 4 (0.54%) |
| Student | 99 (1.06%) | 71 (0.84%) | 36 (0.62%) | 10 (0.22%) | 2 (0.27%) |
| Housewife | 4631 (49.40%) | 4294 (51.06%) | 3215 (55.08%) | 2613 (56.62%) | 398 (53.42%) |
| Unemployed | 615 (6.56%) | 533 (6.34%) | 381 (6.53%) | 316 (6.85%) | 72 (9.66%) |
| Pensioner | 1232 (13.14%) | 1153 (13.71%) | 813 (13.93%) | 715 (15.49%) | 136 (18.26%) |
| Insurance coverage | | | | | |
| No coverage | 572 (6.14%) | 459 (5.49%) | 281 (4.85%) | 203 (4.43%) | 33 (4.44%) |
| Basic package | 6320 (67.79%) | 5605 (67.02%) | 3836 (66.15%) | 2949 (64.33%) | 445 (59.81%) |
| Complementary package | 2431 (26.08%) | 2299 (27.49%) | 1682 (29.01%) | 1432 (31.24%) | 266 (35.75%) |
| | Hypertensive patients (n = 9420) | Screened (n = 8451) | Diagnosed (n = 5866) | Treated (n = 4643) | Controlled (n = 747) |
| **Comorbidities** | | | | | |
| Cardiovascular disease | 328 (3.49%) | 319 (3.79%) | 279 (4.78%) | 268 (5.80%) | 67 (8.97%) |
| Diabetes mellitus | 1621 (23.56%) | 1554 (24.96%) | 1228 (28.26%) | 1092 (31.91%) | 182 (32.79%) |
| Smoking | 2031 (21.65%) | 1815 (21.56%) | 1209 (20.72%) | 921 (19.96%) | 173 (23.16%) |
| Dyslipidemia | 2947 (31.37%) | 2854 (33.88%) | 2316 (39.65%) | 1969 (42.64%) | 343 (45.92%) |
| Body mass index, kg/m$^2$ | | | | | |

*(Continued)*

**Table 1.** (Continued)

| | Hypertensive patients (n = 9420) | Screened (n = 8451) | Diagnosed (n = 5866) | Treated (n = 4643) | Controlled (n = 747) |
|---|---|---|---|---|---|
| <17.5 | 85 (0.93%) | 67 (0.82%) | 44 (0.78%) | 24 (0.54%) | 6 (0.84%) |
| [17.5–25] | 2279 (24.95%) | 1982 (24.21%) | 1304 (23.10%) | 982 (22.04%) | 189 (26.51%) |
| [25–30] | 3615 (39.57%) | 3239 (39.57%) | 2204 (39.04%) | 1754 (39.36%) | 287 (40.25%) |
| [30–35] | 2238 (24.50%) | 2059 (25.15%) | 1468 (26.01%) | 1184 (26.57%) | 158 (22.16%) |
| ≥35 | 918 (10.05%) | 839 (10.25%) | 625 (11.07%) | 512 (11.49%) | 73 (10.24%) |

Data are reported as number (percentage).

4). This association was observed across all age groups and was not limited to the elderly or the very young individuals. Notably, the age disparity in hypertension care, with younger patients being less likely to receive appropriate care, was more pronounced in rural than urban areas, as the gap between age groups was wider in rural communities for all steps of care (S1-S4 Figs in S1 File). Another important feature appearing in the top six in all steps of care was sex. Female sex was associated with a higher probability of being screened (MPP: 0.91 versus 0.86), diagnosed (MPP: 0.66 versus 0.56), treated (MPP: 0.52 versus 0.47), and achieving control (MPP: 0.10 versus 0.09) for hypertension compared to males.

The level of education had a varying association with receiving care in different steps. Higher education attainment was associated with a higher likelihood of being screened for hypertension (MPP in ascending order of education attainment: 0.87, 0.88, and 0.89; Fig 1) and achieving BP control (MPP in ascending order of education attainment: 0.09, 0.10, and 0.11; Fig 4). Conversely, a lower level of education was associated with a better chance of being

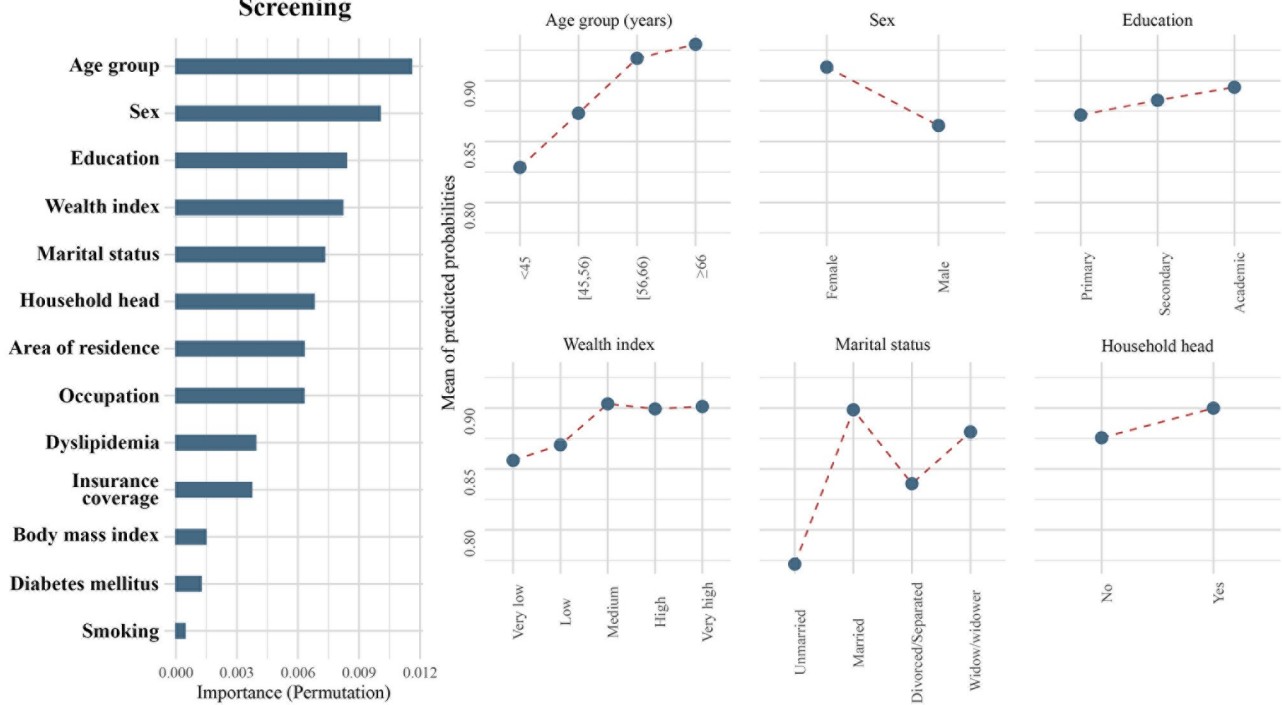

**Fig 1. Importance of population characteristics and comparative probabilities of the top six important classifiers for screening.**

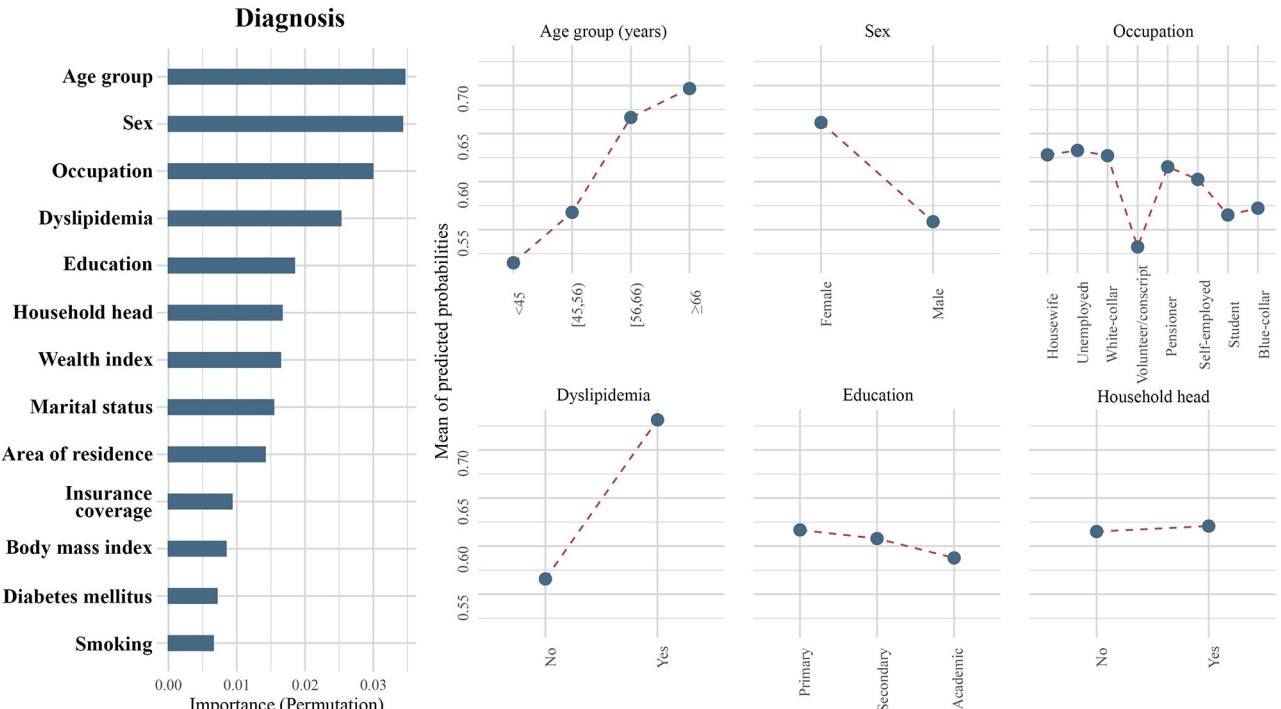

**Fig 2. Importance of population characteristics and comparative probabilities of the top six important classifiers for diagnosis.**

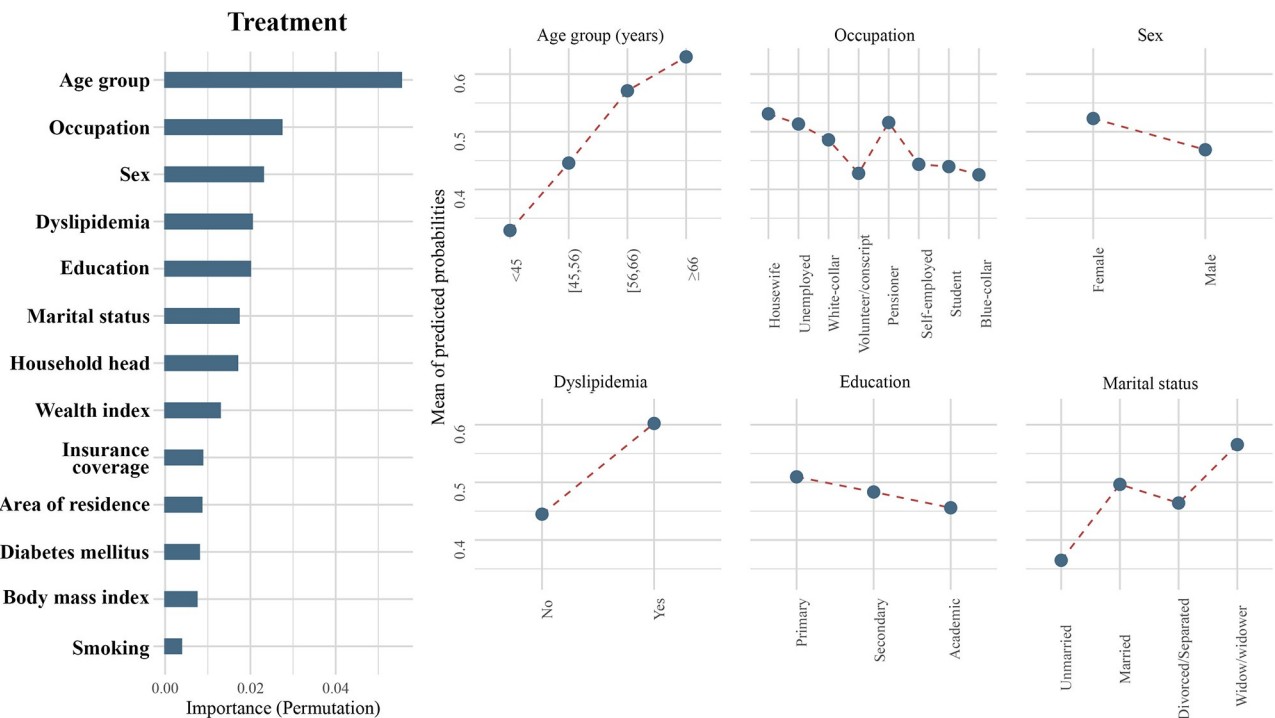

**Fig 3. Importance of population characteristics and comparative probabilities of the top six important classifiers for treatment.**

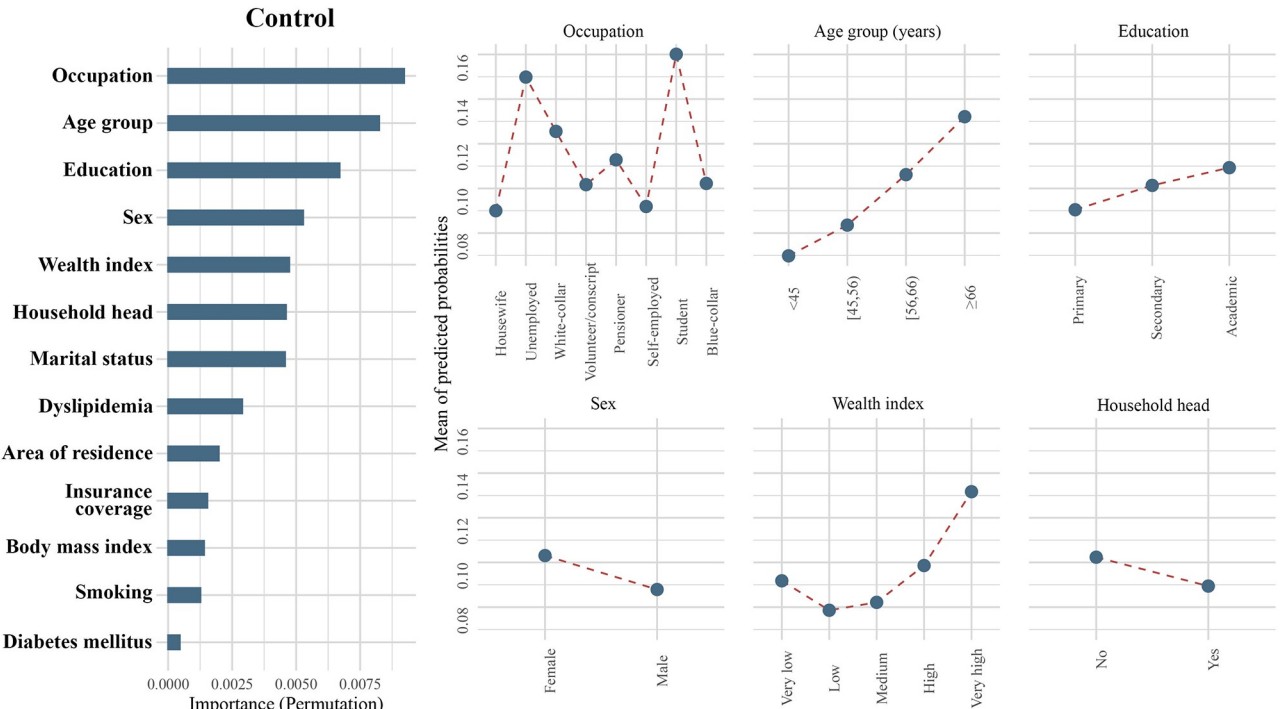

**Fig 4. Importance of population characteristics and comparative probabilities of the top six important classifiers for control.**

diagnosed (MPP in ascending order of education attainment: 0.62, 0.61, and 0.59; Fig 2) and treated (MPP in ascending order of education attainment: 0.51, 0.48, and 0.46; Fig 3).

Wealth index appeared in the top six associated features at the level of screening and control. Individuals with very low (MPP = 0.86) and low (MPP = 0.87) wealth indices had a lower likelihood to be screened for hypertension. Medium (MPP = 0.90), high (MPP = 0.90), and very high (MPP = 0.90) wealth groups were similar in terms of hypertension screening (Fig 1). For hypertension control, a higher wealth index was associated with a better outcome. (MPP in ascending order of wealth index: 0.09, 0.08, 0.08, 0.10, and 0.13; Fig 4). While a higher wealth index meant a higher probability of receiving care, wealth showed interactions with education and area of residence. Higher wealth did not result in enhanced screening, diagnosis, or treatment among individuals with only primary educational attainment, contrary to better educated individuals (S5-S12 Figs in S1 File). Among individuals with low wealth indices, those living in rural areas received better care compared to urban communities; however, among individuals with higher levels of wealth, more appropriate care was observed in urban areas (S5-S8 Figs in S1 File). This trend was specially observed for screening and diagnosis. Notably, urban communities had a higher likelihood of being screened, diagnosed, or treated, but a lower rate of achieving control compared to patients living in rural areas (S1-S8 Figs in S1 File).

Marital status was an important factor in determining if individuals reached screening and treatment (Figs 1 and 3). Single/unmarried people had by far the lowest probability of being screened (MPP = 0.77) or treated (MPP = 0.36) for hypertension. For both screening and treatment steps, divorced/separated patients (MPP for screening = 0.84; and treatment = 0.46) appeared as the second most vulnerable group. On the other hand, married patients (MPP for screening = 0.90; and treatment = 0.50), and widows/widowers (MPP for screening = 0.88; and treatment = 0.57) had a better chance of being screened or treated. Investigating the interactions of marital status with age, sex, area of residence, and wealth showed that while females

generally had better outcomes than males, an exception to this trend was observed among single individuals, as single males had a higher likelihood to be appropriately diagnosed and treated than single females (S13-S20 Figs in S1 File). Marital status did not show a consistent interaction with other variables.

Occupation showed a strong, yet heterogeneous association with reaching steps. For the diagnosis step, volunteers/military conscripts had the lowest probability of receiving care (MPP = 0.53), followed by students (MPP = 0.57), blue-collar workers (MPP = 0.57), and self-employed individuals (MPP = 0.60). Hypertensive pensioners had a higher chance of being diagnosed (MPP = 0.62), while white-collar clerks (MPP = 0.63), unemployed persons (MPP = 0.63), and housewives (MPP = 0.63) were the most likely to receive appropriate hypertension diagnosis (Fig 2). In the treatment step, blue-collar workers (MPP = 0.43) and volunteers/military conscripts (MPP = 0.43) demonstrated the lowest probability of being cared for. In ascending order, students (MPP = 0.44), self-employed workers (MPP = 0.44), white-collar clerks (MPP = 0.49), unemployed individuals (MPP = 0.51), pensioners (MPP = 0.52), and housewives (MPP = 0.53) had a higher chance of being treated for their diagnosed hypertension (Fig 3). On the other hand, in the BP control step, being self-employed (MPP = 0.09) or a housewife (MPP = 0.09) was associated with the lowest probability of achieving BP targets. The next most ineffective hypertension control was observed in blue-collar workers (MPP = 0.10) and volunteers/military conscripts (MPP = 0.10). Pensioners (MPP = 0.11) and white-collar clerks (MPP = 0.13) had a better chance for reaching their BP goal. Control was most successful among students (MPP = 0.16), followed by unemployed individuals (MPP = 0.15; Fig 4). Being the head of household was another socio-economic feature among top classifiers in the screening, diagnosis, and control steps (Figs 1, 2 and 4). The heads of households had a higher chance of being screened (MPP: 0.90 versus 0.88). On the contrary, being the head of household was associated with a lower likelihood than other family members for achieving BP control (MPP: 0.09 versus 0.10).

The only cardiovascular comorbidity appearing in the top six associated features was dyslipidemia. Among hypertensive patients, the presence of dyslipidemia was associated with a higher chance of receiving the appropriate hypertension diagnosis (MPP: 0.73 versus 0.57) and treatment (MPP: 0.60 versus 0.40; Figs 2 and 4). Investigation of interactions showed that among younger age groups, underweight and normal-weight individuals had a lower probability to be screened, diagnosed or treated. In the control step, however, a higher BMI was associated with lower achievement of BP targets. Importantly, smoking and diabetes did not show a meaningful association with receiving care in any sub-group of the population (S21-S28 Figs in S1 File).

## Discussion

This nationally representative data implies that there is substantial room for improvement in the care coverage for hypertension in Iran. While almost nine out of ten hypertensive patients had had BP screening, two thirds had received the appropriate diagnosis by an HCP, only half had been treated with anti-hypertensive medications prior to the survey, and about 8% had achieved BP control.

Our results from the Iranian health-care performance were comparable to other LMICs. According to a 2019 analysis of pooled individual-level data from 44 LMICs (not including Iran), 74% of hypertensive patients had received screening, 39% had a prior diagnosis of hypertension, 30% had been treated, and only 10% had proper BP control; however, these numbers had large variations among countries [8]. A recent systematic review of hypertension care in Arab countries concluded that more than 40% of all hypertensive patients were unaware of their condition, while less than 21% were left untreated [23]. Compared to high-

income countries, the hypertension care coverage in Iran and other LMICs appears to be lower, especially in the control step. A study of near half a million individuals from 12 high-income countries showed that the proportion of awareness (defined as having received the diagnosis of hypertension) was 56–87% and 46–84%, treatment was 55–80% and 39–81%, and control was 26–58% and 17–69%, among women and men, respectively [24].

In our study, the high rate of screening seems encouraging, especially when compared to other LMICs [8]; however, screening did not lead to proper diagnosis, treatment, and control. A 2005 study with similar definitions of hypertension, diagnosis, and treatment reported a diagnosis rate of 49.2% and a treatment rate of 35.7% among Iranian individuals with hypertension [25]. In comparison, our data showed better coverage in the diagnosis and treatment steps which probably indicates improvement in hypertension care between 2005 and 2016. On the other hand, control rates remained low in our study. Importantly, while hypertension can be controlled by oral medications, there are myriad other factors that influence BP levels, such as dietary habits, physical activity, environmental risk factors like air pollution [26, 27], adherence to medications, and continuation of visits with the same provider [28]. Among the Iranian population, consumption of salt is higher than the recommended amounts [29], and almost half of adults have an insufficient level of physical activity [30]. To some extent, these observations might explain the failure to achieve BP control among Iranians, which indicates a need for implementing population-level strategies and health education to modify lifestyle.

We employed ensemble learning, as a superior approach to conventional regression models [17], for analysis of care cascade to find the characteristics associated with hypertension care coverage. Machine learning methods provide many advantages over conventional statistical models in interpreting large datasets [10]. One aspect of random forest model is that it essentially examines the effects of all variables in the dataset simultaneously in deciding the outcome. Incorporation of potential interactions in the model eliminates the possibility of confounding among included variables. We found that among hypertensive patients, younger age, male sex, being unmarried or divorced, lower wealth, or having certain vulnerable occupations were features consistently associated with a lower probability of receiving care. These findings can inform and facilitate future policies to address the existing gaps in hypertension care. By identifying groups who are more likely to be missed at each level, efforts can be made to include more vulnerable individuals in the cascade of care and ultimately, prevent downstream end-organ damage and cardiovascular events attributable to high BP [31].

According to our results, there should be a particular focus on younger adults with hypertension, among whom high BP was more likely to be missed in all steps of care. Importantly, we observed that a young adult who is not overweight or obese, i.e., not the stereotype of a hypertensive patient, was more likely to be neglected for screening, diagnosis, and treatment of hypertension. This suggests that younger individuals may underestimate the risk associated with hypertension, and the health-care system may direct fewer resources to NCDs prevention in young adults. Moreover, the age-related gap in hypertension care cascade was wider in rural compared to urban communities, which could be due to lower health information access and health literacy in rural areas [32]. The higher probability for women to receive hypertension care was compatible with findings from previous studies [8, 13, 24, 25]. The reasons for this observation are multiple, and may include gender differences in health-care-seeking behaviors [33], or a higher emphasis on BP management resulting from perinatal care.

In this study, individuals with lower wealth were less likely to reach higher stages. Although anti-hypertensive drugs are both affordable and accessible in Iran, an analysis of the Iranian Food and Drug Administration data in 2002–2011 demonstrated a wealth-related inequality regarding the use of anti-hypertensive medications among provinces [34]. This evidence explains the important role of wealth index observed in achieving BP control, and underlines the priority

of developing accessible prevention strategies in LMICs. Importantly, a high wealth-index did not translate into better care among the group with low levels of educational attainment, highlighting the critical and intertwined role of socio-economic features in hypertension care.

Among comorbidities, dyslipidemia was associated with a higher probability of being diagnosed and treated; however, history of cardiovascular disease, diabetes mellitus, smoking, or obesity did not appear among the top classifiers. This observation may be concerning as it means that patients with comorbidities, who are at higher risk, were not prioritized for reaching BP control. Future policies should ensure that higher risk groups remain in the care cascade for an integrated risk factor management.

This study provides insight into the current state of hypertension care at the national level. Use of a nationally representative data encompassing a broad range of individual characteristics can be regarded as a strength of this study. A central feature is the use of machine learning for evaluation of hypertension care cascade, which can inform future policies by identifying the characteristics that are predictors of being lost to care at different steps. This study has several limitations. First, the cross-sectional design of STEPs limits our evaluation of the care cascade. While the other available studies have similarly conducted care cascade analyses on cross-sectional data [8, 9, 13, 24], longitudinal studies can provide more accurate results. Second, we included a previous diagnosis by an HCP in our definition of hypertension. There is a possibility that some of the patients who reported a previous hypertension diagnosis, had a BP < 140/90 mmHg, and did not receive treatment, were not actually hypertensive. This might have led to the inclusion of normotensive patients and underestimation of treatment and control rates; however, the number of such normotensive patients was expected to be low, and we chose this design to improve the sensitivity for detecting hypertension in the study population. Third, for a reliable BP reading, measurement should be performed in more than one occasion, and ideally with out-of-office techniques; however, due to the limitations in the design of STEPs, we could only use measurements from one patient encounter. Fourth, we selected conventionally used BP thresholds for hypertension. It should be noted that these thresholds are not in complete agreement with recommendations of most recent hypertension guidelines [35], and using different thresholds may lead to changes in results. Lastly, we could not develop a predictive model for hypertension care based on the available data. Future studies, such as future STEPs surveys in Iran, could be used for this purpose.

## Conclusion

Data from the nationally representative Iranian STEPs survey showed that hypertension care in the country is mostly missing hypertensive individuals in the treatment and control stages. A random forest model determined features associated with hypertension care and indicated targets for improvement. The most important observations were that younger adults, especially those living in rural areas or without conventional hypertension risk factors such as obesity, were more likely to miss care cascade steps. Moreover, males generally had a lower state of care compared to females. Other important features associated with lower care coverage were low wealth, unmarried or divorced status, or occupations such as being a blue-collar worker or self-employment. Random forest model is a helpful tool for recognizing patterns of care coverage for NCDs and their risk factors.

## Supporting information

**S1 File.**
(PDF)

## Acknowledgments

The authors would like to express their appreciation for partnership of deputy for public health and deputy for research and technology of the Ministry of Health and Medical Education, Islamic Republic of Iran's, National Institute for Health Research, and many scholars and experts in related fields.

## Author Contributions

**Conceptualization:** Hamed Tavolinejad, Negar Rezaei, Farshad Farzadfar.

**Data curation:** Shahin Roshani, Erfan Ghasemi.

**Formal analysis:** Hamed Tavolinejad, Shahin Roshani.

**Funding acquisition:** Farshad Farzadfar.

**Investigation:** Mohammad-Mahdi Rashidi.

**Methodology:** Hamed Tavolinejad, Shahin Roshani, Erfan Ghasemi, Moein Yoosefi, Nazila Rezaei, Sina Azadnajafabad, Mohammad-Reza Malekpour.

**Project administration:** Nazila Rezaei, Mohammad-Mahdi Rashidi.

**Resources:** Nazila Rezaei.

**Software:** Shahin Roshani.

**Supervision:** Negar Rezaei.

**Validation:** Erfan Ghasemi, Moein Yoosefi, Azin Ghamari.

**Visualization:** Shahin Roshani.

**Writing – original draft:** Hamed Tavolinejad.

**Writing – review & editing:** Negar Rezaei, Azin Ghamari, Sarvenaz Shahin, Sina Azadnajafabad, Mohammad-Reza Malekpour, Mohammad-Mahdi Rashidi, Farshad Farzadfar.

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
