## [Decision Letter · Decision Letter 0]

3 Dec 2021

PONE-D-21-32708A machine learning approach to evaluate the state of hypertension care coverage: From 2016 STEPs survey in IranPLOS ONE

Dear Dr. Farzadfar,

Thank you for submitting your manuscript to PLOS ONE. After careful consideration, we feel that it has  some merit but does not fully meet PLOS ONE’s publication criteria as it currently stands. Therefore, we invite you to submit a revised version of the manuscript that addresses the points raised during the review process. Specifically, please address all comments made by the reviewers. Be sure to:Indicate what is the added value of the study and its contribution to the fieldAddress all concern in its method (design/analysis) Please submit your revised manuscript by Jan 17 2022 11:59PM. If you will need more time than this to complete your revisions, please reply to this message or contact the journal office at plosone@plos.org. Please include the following items when submitting your revised manuscript:A rebuttal letter that responds to each point raised by the academic editor and reviewer(s). You should upload this letter as a separate file labeled 'Response to Reviewers'.A marked-up copy of your manuscript that highlights changes made to the original version. You should upload this as a separate file labeled 'Revised Manuscript with Track Changes'.An unmarked version of your revised paper without tracked changes. You should upload this as a separate file labeled 'Manuscript'.

We look forward to receiving your revised manuscript.

Kind regards,

Amir Radfar, MD,MPH,MSc,DHSc

Academic Editor

PLOS ONE

Journal Requirements:

Reviewers' comments:

Reviewer's Responses to Questions

**Comments to the Author**

1. Is the manuscript technically sound, and do the data support the conclusions?

Reviewer #1: Partly

Reviewer #2: No

Reviewer #3: Partly

2. Has the statistical analysis been performed appropriately and rigorously? 

Reviewer #1: Yes

Reviewer #2: No

Reviewer #3: N/A

3. Have the authors made all data underlying the findings in their manuscript fully available?

Reviewer #1: Yes

Reviewer #2: No

Reviewer #3: Yes

4. Is the manuscript presented in an intelligible fashion and written in standard English?

Reviewer #1: Yes

Reviewer #2: Yes

Reviewer #3: Yes

5. Review Comments to the Author

Reviewer #1: The publication "A machine learning approach to evaluate the state of hypertension care coverage: From 2016 STEPs survey in Iran" by Tavolinejad et al. tackles an important problem of development and progression of hypertension, moreover it is driven by the low- to middle-income country data. The authors consider hypertension to be defined as blood pressure ≥ 140/90 mmHg), or reported use of anti-hypertensive medications, or a previous hypertension diagnosis. The four steps of care considered for the analysis included screening, diagnosis, treatment, or control. Using machine learning approach (random forest), the authors estimated that from 30.8% of population with hypertension, 89.7% were screened, 62.3% received diagnosis, 49.3 were treated, and 7.9% achieved control. The random forest analysis indicated that younger age, male sex, low wealth, and unmarried/divorced status were associated with a lower probability of receiving care. As this work represents a high educational value to the populations affected by hypertension as well as may serve a the learning about detection and consequences early, this potential is diminished by several major issues related to the following aspects:

1. Although the statistical data is rich and presented clearly (Table 1), the figures in the main manuscript are not readable. On the other hand, the figures in the supplementary files are clear and of high quality. The analyses are performed with a good technical standard and are described in sufficient detail however I disagree with the usage of the word "novel" toward the random forest application in these population data.

2. The authors used random forest to evaluate the features importance for each of the stages, while the post analyses of random forest include other valuable applications that I highly recommend to the authors. To do that, the authors should move beyond feature importance or probability distribution within. The study will benefit the audience if the authors consider:

- plotting/analyzing the probability data in higher dimension to seek for hidden relationships and patterns between e.g., age, education, status, gender, etc, i.e., patterns not limited to the conclusions that are expected from linear analysis of individual data (young single male with low income having lower care). Using the four figures to support the findings seems as a very good start to more advanced analyses and more informative conclusions. In addition, potential relationships with the biological class of features (BMI, diabetes, smoking) would start showing to play a role in only a particular cohort.

- how do the classifiers cluster (clustering) or relate (PCA) to one another among population?

- building a predictive model from the data, perhaps for the remaining large cohort of the population not associated with hypertension.

3. It is unclear if step "screening" is associated with HP when introduced in the abstract.

4. Lines 90-91: is condition (I) considered when "SBP and DBP" or "SBP or DBP" are present?

5. Line 99: the term "screening" still doesn't explain clearly if the BP measured was ever related to HP.

6. Line 117: it is unclear what authors mean by "Random forest is an ensemble model that handles categorical variables AS THEY ARE"?

7. Lines 118-119: authors should expand on: "...it provides appropriate and competitive predictive accuracy compared to other algorithms". It is recommended that authors make an explicit comparison of their analyses with other machine learning approaches before stating that random forest is the best choice for it.

8. Line 119: definition of "hyper-parameters" should be put early for the readers.

9. Line 131: it is unclear what authors mean by knowledge in: "...tools were used to extract knowledge".

10. Line 161 and further: the class of "white collar" should be specified as either clerks or workers for consistency.

11. Lines 184-185: The sentence "Very low (MPP=0.86) and low (MPP=0.87) wealth indices were associated with a decreased probability of being screened." is unclear.

12. It is suggested that, given the longitudinal data/steps present, the authors should analyze the longitudinal data on additional level; one idea could be to look at the trends in features' importance while moving along the 4 steps. More informative analysis could include population models for the subpopulations pre-classified with the machine learning tools.

In summary, the study should be enhanced by computational analyses and graphical representation of the results, including, yet not limited to those suggested in #2 and #12. If these additional analyses are added, the manuscript could be submitted to the follow-up review.

Reviewer #2: The manuscript presents a Random Forest based machine learning approach to evaluate the state of hypertension care coverage through a population-based data from Iran. Unfortunately, the paper falls short in bringing any fresh insights to the readers. Most of the causes and recommendations made by the authors are already known to the community, and it is unclear what new insight machine learning is offering, if any. Bulk of the paper is dedicated to reporting what the tool running the Random Forest model outputs, without adequate deeper dive. There is little detail on handling the data itself, as what was done to pre-process, normalize and standardize the data prior to feeding it to the machine learning model. Given the understandable lack of availability of the dataset in public domain, it is useful to create credibility around the analysis through rigorous statistics and figures. The paper lacks to explain as what the predictions mean in context of the data as well as the field itself. On what could be a very valuable dataset to analyse, I am afraid that there is a need for extended analysis and deeper interpretation of results, both in terms of machine learning model as well as for the hypertension care coverage.

Reviewer #3: The authors present a very well-written and important paper that helps assess the reasons for the discontinuation in the care pathway through the diagnosis, treatment and control of hypertension in Iran. For that they use the random forest algorithm and conclude that younger adults, men, unmarried or divorced people and those with lower socio-economical status are more at risk to not receive treatment and achieve control of their blood pressure.

Overall the paper was clear and presented the findings in an understandable fashion. I have two main suggestions to be discussed with the authors in two different domains.

(A) Guideline Selection and Phenotyping Algorithm

1. Firstly, I would like to understand the reason for the selection of the older AHA guideline (2017) since in 2020 there was the release of a new version. The new guideline document can be found in the following weblink: https://www.ahajournals.org/doi/10.1161/HYPERTENSIONAHA.120.15026. In it, there are some different recommendations for the essential and optimal control of blood pressure.

2. Also, according to the author's phenotyping algorithm, hypertensive patients had SBP >= 140 and DBP >= 90 and controlled hypertensive SBP < 130 and DBP < 80. This leaves a gap in the participants who had SBP in between 130-139 and 80-89 (the new guideline covers that gap). I suggest that a closer look is taken into that.

3. Again, regarding the guideline, usually blood pressure is measured in more than one encounter or with out-of-office techniques to discard white coat and masked hypertension. I do understand the cross-section nature of the STEPs study and the lack of longitudinal data. However, I believe this should be mentioned in the limitations as well.

(B) Algorithm Selection and Hyper-Parameter Tuning

1. The use of the random forest algorithm is quite interesting and it is indeed powerful. However, there is another class of ensemble algorithms that usually outperforms random forest called gradient boosting machines (examples are XGBoost and LightGBM) which generally use decision trees. In that aspect, I would like to hear from the authors the reason for opting for random forest instead of one of those algorithms.

2. Lastly, it was not clear to me how the hyper-parameter tuning was executed. This is a very important step and it is good practice to have a separated validation set for that. It would be beneficial to understand the methodology used by the authors and have it thoroughly explained in the paper or in the supplement for better judgment of the results and also increased reproducibility.

6. PLOS authors have the option to publish the peer review history of their article (what does this mean?). If published, this will include your full peer review and any attached files.

Reviewer #1: No

Reviewer #2: No

Reviewer #3: **Yes: **Ariane Sasso

---

## [Author Response · Author response to Decision Letter 0]

5 Mar 2022

Dear Professor Radfar, 

Academic editor of the PLOS ONE,

We are pleased to resubmit our revised manuscript entitled “A machine learning approach to evaluate the state of hypertension care coverage: From 2016 STEPs survey in Iran” (Manuscript ID: PONE-D-21-32708). On behalf of our research team, I would like to thank the respected editorial office of PLOS ONE and the honorable reviewers for the time and effort they took in evaluating our work. We carefully addressed the insightful comments of the reviewers. The following pages include a point-by-point response to the comments with references to the changes made in the manuscript. We made sure to indicate the added value and contribution of this study, which is an in-depth analysis of the state of hypertension care in Iran—a relevant topic considering the paucity of data from LMICs in this regard. Moreover, we addressed the concerns raised about the methodology of this investigation. As instructed, we have submitted a revised version with changes marked in highlight and a second copy without any markings. We have applied PLOS ONE style requirements to our files. We declare that this study was funded by Ministry of Health and Medical Education and National Institute for Health Research (grant number: 241-93259), in the financial disclosure and funding information.

Regarding our data availability statement, it should be noted that the STEPS 2016 study is a project launched by the Iranian Ministry of Health and Medical Education of Iran (MOHME), which owns the rights to the dataset. However, interested and qualified researchers may contact the Non-Communicable Diseases Research Center (www.ncdrc.net) to access the datasets of the STEPS 2016 study. The aggregated level data are freely accessible via https://vizit.report/en/index.html. The author-generated codes can be uploaded to GitHub and be made freely available, if the journal considers this source suitable. Otherwise, we are ready to upload the code in alternative sources that the journal would suggest. 

We are looking forward to receiving your kind response and decision.

Farshad Farzadfar (MD, MPH, MSc, DSc)

Non-Communicable Diseases Research Center, 

Endocrinology and Metabolism Population Sciences Institute, 

Tehran University of Medical Sciences, Tehran, Iran 

Address: Second Floor, No.10, Jalal Al-e-Ahmad Highway, Tehran, Iran

Postal Code: 1411713137

Tel: +98-21-88631293 

Email: f–farzadfar@tums.ac.ir

---

## [Decision Letter · Decision Letter 1]

7 Jun 2022

PONE-D-21-32708R1A machine learning approach to evaluate the state of hypertension care coverage: From 2016 STEPs survey in IranPLOS ONE

Dear Dr. Farzadfar,

Thank you for submitting your manuscript to PLOS ONE. After careful consideration, we feel that it has merit but does not fully meet PLOS ONE’s publication criteria as it currently stands. Therefore, we invite you to submit a revised version of the manuscript that addresses the points raised during the review process.

 Specifically,please address point B from the reviewer #3 which is still missing.

We look forward to receiving your revised manuscript.

Kind regards,

Amir Radfar, MD,MPH,MSc,DHSc

Academic Editor

PLOS ONE

Journal Requirements:

Reviewers' comments:

Reviewer's Responses to Questions

**Comments to the Author**

1. If the authors have adequately addressed your comments raised in a previous round of review and you feel that this manuscript is now acceptable for publication, you may indicate that here to bypass the “Comments to the Author” section, enter your conflict of interest statement in the “Confidential to Editor” section, and submit your "Accept" recommendation.

Reviewer #1: All comments have been addressed

Reviewer #3: (No Response)

2. Is the manuscript technically sound, and do the data support the conclusions?

Reviewer #1: Yes

Reviewer #3: Yes

3. Has the statistical analysis been performed appropriately and rigorously? 

Reviewer #1: Yes

Reviewer #3: Yes

4. Have the authors made all data underlying the findings in their manuscript fully available?

Reviewer #1: Yes

Reviewer #3: No

5. Is the manuscript presented in an intelligible fashion and written in standard English?

Reviewer #1: Yes

Reviewer #3: Yes

6. Review Comments to the Author

Reviewer #1: Thanks to the Authors for addressing all the comments and explaining the unclear statements. The article is now recommended for publishing.

Reviewer #3: I have one last point that needs addressing. Overall the paper was clear and presented the findings in an understandable fashion. Most of the recommendations from the previous review were followed.

1) Guideline Selection and Phenotyping Algorithm

Items A and C were answered satisfactorily. However, point B is still missing:

B. Also, according to the author's phenotyping algorithm, hypertensive patients had SBP >= 140 and DBP >= 90 and controlled hypertensive SBP < 130 and DBP < 80. This leaves a gap in the participants who had SBP between 130-139 and 80-89 (the new guideline covers that gap). I suggest that a closer look is taken into that (check lines 91-95).

2) Algorithm Selection and Hyper-Parameter Tuning

All the items were answered satisfactorily. In the future, the implementation of algorithms such as XGBoost and LightGBM is currently as easy as RF (there are open-source packages and libraries in Python), and they could potentially be tried out.

7. PLOS authors have the option to publish the peer review history of their article (what does this mean?). If published, this will include your full peer review and any attached files.

Reviewer #1: No

Reviewer #3: **Yes: **Ariane Sasso

---

## [Author Response · Author response to Decision Letter 1]

20 Jul 2022

Dear Professor Radfar, 

Academic editor of the PLOS ONE,

We are pleased to resubmit our revised manuscript entitled “A machine learning approach to evaluate the state of hypertension care coverage: From 2016 STEPs survey in Iran” (Manuscript ID: PONE-D-21-32708R1). On behalf of my colleagues, I would like to thank the esteemed editors and staff of PLOS ONE for consideration of our work. As instructed, we have submitted a revised version with tracked changes and a second copy without any markings. The following pages include a point-by-point response to the comments.

In response to the insightful comment of Reviewer #3, we have explained that the selected blood pressure thresholds were used to best reflect the state of care in Iran at the time the STEPs survey was performed. Moreover, these cut-offs are widely used in other low- and middle-income countries (LMICs), and are comparable to other available data from LMICs. We note the trade-off in selecting thresholds, which would lead to underestimation or overestimation of hypertension prevalence, diagnosis, treatment, and control. We have tried to explain our rationale more clearly, and we note in the manuscript that these thresholds may not be in total agreement with the most recent guidelines. We would appreciate any specific recommendation about changing blood pressure thresholds that could improve our paper.

Please kindly note a change we have made to the author affiliations in this revision. Furthermore, we have added two references (numbers 15 and 35) to the manuscript. As you instructed, we have checked and ensured that our references list is complete and correct. The journal requirements also stated “If you have cited papers that have been retracted, please include the rationale for doing so in the manuscript text, or remove these references and replace them with relevant current references”. We are not sure if this was an automated message or was in reference to a specific issue with our manuscript. We did not find a retracted article in our reference list, but please kindly let us know if there is a reference we should recheck and address. 

Please do not hesitate to contact me if we need to provide any additional information about this submission. We are looking forward to receiving your kind response and decision.

Farshad Farzadfar (MD, MPH, MSc, DSc)

Non-Communicable Diseases Research Center, 

Endocrinology and Metabolism Population Sciences Institute, 

Tehran University of Medical Sciences, Tehran, Iran 

Address: Second Floor, No.10, Jalal Al-e-Ahmad Highway, Tehran, Iran

Postal Code: 1411713137

Tel: +98-21-88631293 

Email: f–farzadfar@tums.ac.ir

 

Reviewer #1:

“Thanks to the Authors for addressing all the comments and explaining the unclear statements. The article is now recommended for publishing.”

Response: Dear respected reviewer, we thank you for your time and your constructive comments, which have improved our paper.

Reviewer #3:

“I have one last point that needs addressing. Overall the paper was clear and presented the findings in an understandable fashion. Most of the recommendations from the previous review were followed.

“1) Guideline Selection and Phenotyping Algorithm

“Items A and C were answered satisfactorily. However, point B is still missing:

“B. Also, according to the author's phenotyping algorithm, hypertensive patients had SBP >= 140 and DBP >= 90 and controlled hypertensive SBP < 130 and DBP < 80. This leaves a gap in the participants who had SBP between 130-139 and 80-89 (the new guideline covers that gap). I suggest that a closer look is taken into that (check lines 91-95).”

Response: Dear respected reviewer, we thank you for your precise and constructive comment. We apologize for we did not appropriately address one of your comments in the previous letter.

Your point is completely accurate. There is a gap in blood pressure values for definition of hypertension and control, which can be eliminated by reducing the BP cut-off for defining hypertension/high-normal BP to ≥130/80, or even by increasing the target BP for control to <140/90. 

The main reason we have chosen the 140/90 threshold for hypertension is we wanted to evaluate health-care system coverage at the time of the study. We had reservations about using a lower threshold, since we were concerned it would not make an accurate simulation of the state of care and would not provide a reliable estimation of the factors associated with care, as most health-care providers may not have labeled hypertension and considered pharmacological treatment for the group with office measurements at 130-139/80-89 (high-normal BP). Our rationale was that this would underestimate the rate of treatment and control. Furthermore, the conventional cut-off of 140/90 is widely used in other low- and middle-income countries—as demonstrated by databases used in the following study: https://doi.org/10.1161/CIRCULATIONAHA.120.051620.

Importantly, it should be noted that having a BP measurement ≥140/90 was not our only criteria for hypertension definition. We also used hypertension diagnosis by health-care provider, or drug-history of BP-lowering medications. Therefore, a patient with BP<140/90 who had one of the above criteria could still be considered in the hypertensive population. Such patients would then be classified as achieving control if they had fulfilled both diagnosis and treatment steps, and had BP <130/80.

As for the control cut-off, we used 130/80 as the target because it is an optimum value for most patients for reducing adverse outcomes, and it is recommended by recent guidelines. 

Ultimately, we understand that choosing any cut-off for definition of hypertension and for control would result in specific limitations. Especially since we do not have longitudinal data with multiple measurements (a point that you have thoughtfully mentioned). Thanks to your comment, we have tried to explain our rationale for choosing these numbers more clearly. In brief, these cut-offs were used because they most accurately reflect the state of care in Iran, and in most LMICs. Moreover, as several population-based studies evaluating hypertensive patients in LMICs have used similar cut-offs, this approach would increase the comparability of our results with those of others. We have also added a phrase to limitations to make this clear to readers.

We hope these explanations are satisfactory. Please kindly let us know if you think there is another specific approach that would improve this study. 

“2) Algorithm Selection and Hyper-Parameter Tuning

“All the items were answered satisfactorily. In the future, the implementation of algorithms such as XGBoost and LightGBM is currently as easy as RF (there are open-source packages and libraries in Python), and they could potentially be tried out.”

Response: Many thanks for your insightful and expert comment. This would be a great direction for future studies.

---

## [Decision Letter · Decision Letter 2]

11 Aug 2022

A machine learning approach to evaluate the state of hypertension care coverage: From 2016 STEPs survey in Iran

PONE-D-21-32708R2

Dear Dr. Farzadfar,

We’re pleased to inform you that your manuscript has been judged scientifically suitable for publication and will be formally accepted for publication once it meets all outstanding technical requirements.

Kind regards,

Amir Radfar, MD,MPH,MSc,DHSc

Academic Editor

PLOS ONE

Additional Editor Comments (optional):

Reviewers' comments:

Reviewer's Responses to Questions

**Comments to the Author**

1. If the authors have adequately addressed your comments raised in a previous round of review and you feel that this manuscript is now acceptable for publication, you may indicate that here to bypass the “Comments to the Author” section, enter your conflict of interest statement in the “Confidential to Editor” section, and submit your "Accept" recommendation.

Reviewer #3: All comments have been addressed

Reviewer #4: All comments have been addressed

2. Is the manuscript technically sound, and do the data support the conclusions?

Reviewer #3: Yes

Reviewer #4: Yes

3. Has the statistical analysis been performed appropriately and rigorously? 

Reviewer #3: Yes

Reviewer #4: Yes

4. Have the authors made all data underlying the findings in their manuscript fully available?

Reviewer #3: No

Reviewer #4: Yes

5. Is the manuscript presented in an intelligible fashion and written in standard English?

Reviewer #3: Yes

Reviewer #4: Yes

6. Review Comments to the Author

Reviewer #3: Dear Authors,

Thank you very much for the great work and for addressing all the comments in detail. I am sorry this review process took so long, and I am to blame for not meeting the deadlines as I wished. Hopefully, your paper will be published soon. I wish you much success!

Reviewer #4: Thank you for this well written article. I believe this manuscript will add to the body of literature.

7. PLOS authors have the option to publish the peer review history of their article (what does this mean?). If published, this will include your full peer review and any attached files.

Reviewer #3: **Yes: **Ariane Sasso

Reviewer #4: **Yes: **Irina Filip

---

## [Editor Report · Acceptance letter]

12 Sep 2022

PONE-D-21-32708R2 

A machine learning approach to evaluate the state of hypertension care coverage: From 2016 STEPs survey in Iran 

Dear Dr. Farzadfar:

I'm pleased to inform you that your manuscript has been deemed suitable for publication in PLOS ONE. Congratulations! Your manuscript is now with our production department. 

Kind regards, 

on behalf of

Dr. Amir Radfar 

Academic Editor

PLOS ONE